# Properties of Selenolate-Diselenide Redox Equilibria in View of Their Thiolate-Disulfide Counterparts

**DOI:** 10.3390/antiox12040822

**Published:** 2023-03-28

**Authors:** Tamás Pálla, Arash Mirzahosseini, Béla Noszál

**Affiliations:** 1Department of Pharmaceutical Chemistry, Semmelweis University, Hőgyes Endre utca 9., 1092 Budapest, Hungary; 2Research Group of Drugs of Abuse and Doping Agents, Hungarian Academy of Sciences, Széchenyi István tér 9., 1051 Budapest, Hungary

**Keywords:** selenoproteins, antioxidant capacity, redox potential

## Abstract

Selenium, the multifaceted redox agent, is characterized in terms of oxidation states, with emphasis on selenol and diselenide in proteinogenic compounds. Selenocysteine, selenocystine, selenocysteamine, and selenocystamine are depicted in view of their co-dependent, interfering acid-base, and redox properties. The pH-dependent, apparent (conditional), and pH-independent, highly specific, microscopic forms of the redox equilibrium constants are described. Experimental techniques and evaluation methods for the determination of the equilibrium and redox parameters are discussed, with a focus on nuclear magnetic resonance spectroscopy, which is the prime technique to observe selenium properties in organic compounds. The correlation between redox, acid-base, and NMR parameters is shown in diagrams and tables. The fairly accessible NMR and acid-base parameters are discussed to assess the predictive power of these methods to estimate the site-specific redox properties of selenium-containing moieties in large molecules.

## 1. Introduction

Selenolate-diselenide equilibria are the same as their sulfur-containing counterparts, thiolate-diselenide equilibria, and involve the reversible formation of diselenides from selenolates through the two-electron oxidation of the selenolate group. The latter moiety can also undergo protonation to form selenol. The equilibria between selenolates and diselenides are influenced by a variety of factors, including pH, temperature, and the presence of other biological species such as metal cations and enzymes.

Selenocysteine is a unique amino acid that contains the trace element selenium. It is incorporated into selenoproteins, a class of proteins that perform important biological functions in humans and other organisms. Unlike the synthesis of other amino acids, the synthesis of selenocysteine is a complex process that requires the unique utilization of a stop codon, UGA. The incorporation of selenocysteine into selenoproteins is tightly regulated, and alterations in selenocysteine metabolism or a selenium deficiency have been implicated in several human diseases.

Selenoproteins are found in all domains of life and have been identified in numerous tissues and organs, including the liver, kidney, brain, and immune cells. They are involved in a wide range of physiological processes, such as redox regulation, thyroid hormone metabolism, immune defense, and DNA repair. They may also help prevent cancer, heart disease, and neurological disorders. Despite their importance, the functions of many selenoproteins are still not well understood, and more research is needed to elucidate their roles in various biological processes.

The redox properties of selenolates and diselenides make them important targets for drug development and therapeutic interventions and are of particular interest in the field of selenium biochemistry, as they have important implications for the biological activity of these compounds. Alterations in selenolate-diselenide equilibria have been shown to affect the activity of selenoproteins, which are important mediators of cellular redox signaling and antioxidant defense.

This review will provide an overview of the current understanding of selenium in biomolecules, compared to its sulfur-containing counterparts, then the metabolism of selenium and the role of selenoproteins are discussed, including their antioxidant activity. Next, a detailed physico-chemical discussion of selenolate-diselenide acid-base and redox equilibria is presented, delving into the submolecular protonation and oxidation processes. A case-tailored evaluation method is presented using nuclear magnetic resonance (NMR) techniques to quantify these species-specific parameters. Finally, the relationship between acid-base, redox, and NMR parameters is evaluated as a means to predict and design selenolate-diselenide redox systems.

## 2. Oxidation Stages of Selenium, Comparison with Sulfur

Selenium can be found in scarce quantities in the soil, the oceans, the groundwater, and all biological organisms. Elemental selenium, not accessible directly to biological organisms, occurs in nature in even lower quantities; it barely dissolves in water. Inorganic selenium occurs vastly in one of the following three states, which are distinguished by their oxidation number: −2 (in the form of selenides), +4 (selenous acid and its salts, selenites), +6 (selenic acid and its salts, selenates). Selenates ($SeO_{4}^{2-}$) and selenites ($SeO_{3}^{2-}$) are mostly water-soluble compounds, and therefore account for the majority of selenium in water.

Organoselenium compounds are found in nature mostly in selenol/selenolate, diselenade (the oxidized form of selenolate), and selenide forms. The most important occurrences of selenium in biological organisms are selenomethionine (SeMet) and selenocysteine (SeCys, or CysSeH). However, other forms of organoselenium compounds are also known, with possible oxidation numbers ranging from −2 to +6 (Figure 1).

The physico-chemical and antioxidant characteristics of selenium compounds are commonly compared to those of sulfur analogs. However, sulfur and selenium antioxidant activities can be quite different, with each utilizing different mechanisms to prevent oxidative stress and cellular damage. Figure 2 contains the most important differences between thiol and selenol moieties, as well as some important biological examples of sulfur and selenium compounds [1].

The difference in the redox chemistry of the two elements lies between their two physico-chemical profiles. This difference is substantial for both one-electron and two-electron redox transitions.

One important factor in the biochemically relevant differences is the higher polarizability of Se^−^ and the lower protonation constant of selenolate (or conversely lower p*K*_a_ of SeH). The difference in protonation constants results in a distinct pH-dependence of selenolate antioxidant activity compared to that of thiolates, with selenolates generally being active at a much lower pH [2].

Another important factor is the inability of selenium to generally form π bonds. The outer valence electrons of selenium are more loosely held than those of sulfur. As a result, selenium is a stronger nucleophile and will react with reactive oxygen species faster than sulfur, but the resulting lack of π-bond character in the oxidized form, i.e., Se-O bond, means that the oxidized selenium compound can be much more readily reduced in comparison to oxidized sulfur compounds. This implies that selenium-containing biomolecules are generally more resistant to oxidation compared to their sulfur-containing counterparts [3].

Furthermore, differences in sulfur and selenium chemistry result in a variety of redox states, posttranslational protein modifications containing cysteine or selenocysteine, and metal-free redox pathways. It is common, however, for both elements to participate in several distinct redox pathways, including exchange and radical reactions, as well as atom-, electron-, and hydride-transfer reactions [4].

## 3. Selenium in the Human Body—Selenoproteins from Redox and Acid-Base Perspective

Selenium, as a micronutrient, is essential for living organisms and is largely incorporated into biological molecules in the form of selenocysteine. Selenocysteine is also accounted for as the 21st proteinogenic amino acid and is the sole source of selenium in selenoproteins for humans. Selenoproteins are vitally important macromolecules with an indispensable role in redox homeostasis, fulfilling oxidoreductase enzyme activities in our body.

Selenium is an essential nutrient; however, at higher levels, it becomes toxic. The endogenous synthesis of selenocysteine from inorganic selenium (usually selenide) and serine is encoded by a stop codon UGA in mRNA and involves a unique tRNA. A unique mechanism is used to decode the UGA codon in mRNA to co-translationally incorporate selenocysteine into the growing polypeptide chain since there is no readily available pool of selenocysteine [5]. One of the main sources of our dietary selenium is from plants; however, plants convert Se mainly into selenomethionine. Selenocystine, methylselenocysteine, and c-glutamyl-methylselenocysteine are not significantly incorporated into plant proteins. Higher animals, however, are unable to synthesize selenomethionine. As a result, selenomethionine is nonspecifically incorporated into a wide range of proteins in skeletal muscle. The details of selenium metabolism and storage, particularly of transport and excretion, are limited [6]. It is understood, however, that selenocysteine is considered the most conservative substitution for cysteine in nature. Selenopeptides have very interesting and distinct properties, which lead to a diverse range of applications in synthetic peptide chemistry, such as structural, functional, and mechanistic probes, robust scaffolds, enzymatic reaction design, peptide conjugations, and folding tools [7]. Another unique feature of selenocysteine is the occurrence and properties of its stable and radioactive isotopes. Thus, selenocysteine is particularly useful for residue-specific radiolabeling with gamma or positron emitters, as a reactive handle for electrophilic probes introducing fluorescence or other peptide conjugates, as the basis for affinity purification of recombinant proteins, the trapping of folding intermediates, improved phasing in X-ray crystallography, the introduction of 77Se for NMR spectroscopy, or the analysis or tailoring of enzymatic reactions involving thiolate or selenolate redox chemistry [8].

### 3.1. Selenoproteins in Health and Disease

Selenium’s role in health and disease can be largely attributed to its presence in the approximately 20 identified selenoproteins. Selenoproteins such as glutathione peroxidases, thioredoxin reductases, and iodothyronine deiodinases are involved in redox reactions. Some selenoproteins are also required for sperm motility (mitochondrial capsule selenoprotein) and may reduce the risk of miscarriage. Other selenoproteins include Selenophosphate synthetase, selenoprotein P, selenoprotein W, prostate epithelial selenoprotein, DNA-bound spermatid selenoprotein. Selenoproteins have primary biological functions in oxidoreduction, redox signaling, antioxidant defense, thyroid hormone metabolism, and immune responses. Thus, selenium deficiency is strongly correlated with human diseases such as cancer, Keshan disease, Kashin-Beck disease, virus infections, male infertility, and abnormalities in immune responses and thyroid hormone function [9]. Selenium intake status is also related to health outcomes, in particular gastrointestinal and prostate cancer, cardiovascular disease, and diabetes. While normal physiological selenium levels afford protection against the above diseases compared to selenium deficiency, excessive selenium intake may induce damage to the cardiovascular system and increase the risk of type 2 diabetes [10]. Selenium appears to fulfill a crucial role in the proper functioning of the immune system as well and appears to be a key factor in inhibiting HIV progression to AIDS. Selenium deficiency has also been linked to adverse mood states. Findings have also shown that an elevated selenium intake may be associated with reduced cancer risk [11]. Selenium compounds have been found to inhibit tumor growth as a consequence of selenium intakes improving nutritional deficiencies. It is proposed that while some cancer protection via antioxidant activity involves selenoenzymes, some specific selenium metabolites, which are produced in significant amounts at relatively high selenium intakes, also inhibit antitumorigenic functions. According to this two-stage model, individuals with nutritionally adequate selenium may benefit from selenium supplementation [12].

### 3.2. Selenoprotein Antioxidant Activity

The direct antioxidant actions of selenocysteine in selenoproteins arise from the nucleophilic properties of selenolate (which predominates over the selenol form at physiological pH values) and the ease of oxidation of selenocysteine. This results in higher rate constants and better affinity for oxidation with oxidants compared to the corresponding thiolates. Selenolate-containing proteins may also chelate redox-active metal ions. Selenocysteine in the active site of enzymes broadens and enhances the kinetic properties and catalytic activity of antioxidant enzymes against biological oxidants, compared with cysteine-containing analogs [13].

The first selenoprotein discovered was glutathione peroxidase (EC 1.11.1.9) [14], with four distinct isoforms [15,16,17,18]. Glutathione peroxidase is responsible mainly for the elimination of hydrogen peroxide produced in the oxidation chain reactions. The selenocysteine moiety is essential for its function. The selenolate group reacts directly with peroxides to form a selenenic acid, which is then reduced back to the selenolate form by glutathione with the help of glutathione reductase [19]. One aspect that sheds light on selenium antioxidant mechanisms is the design of some diselenides that show peroxidase activity. These diselenides exert their peroxidase activity via their selenolate form and subsequently as selenenic acid and selenenyl sulfide intermediates, thus resembling the mechanism by which glutathione peroxidase catalyzes the reduction of hydroperoxides [20,21]. It is undoubtedly an important achievement that ebselen has been demonstrated to possess antioxidant and therapeutic properties. The ability of ebselen to mimic the reaction catalyzed by GPx may be viewed as the most important molecular mechanism of action of this class of compound. However, an alternative mechanism was also proposed so that organoselenium compounds can oxidize the thiolate groups of proteins (the thiolate modifier effects of organoselenium compounds) and spare selenoproteins from inactivation by soft electrophiles (CH_3_Hg^+^, Hg^2+^, Cd^2+^) might be another explanation of their pharmacological effects [22].

Iodothyronine deiodinase enzymes (EC 1.21.99.4 and EC 1.21.99.3) are a subfamily of deiodinase enzymes that activate and inactivate thyroid hormones and their precursors. The isoforms I and II of these enzymes catalyze the thyronine-thyroxine conversion, while isoform III inactivates thyroxine [23,24,25,26].

Thioredoxin reductase (EC 1.8.1.9) is the only enzyme capable of reducing thioredoxin. Together, they form the backbone of the thioredoxin system, a main component of antioxidant pathways and regulator of protein disulfide bridges. The selenocysteine of thioredoxin reductase is found in the redox center of the enzyme near its C-terminal [27].

The redox-sensitive nature of the selenolate group makes this class of biomolecules a peculiar component of the redox homeostasis system of our physiology (the important mechanisms involved are depicted in Figure 3). The role of selenolate is so irreplaceable, that substituting selenocysteine in these enzymes for cysteine results in a dramatic loss of enzymatic activity [19,23]. The oxidized form of selenocysteine, produced by a two-electron oxidation step, is selenocystine (CysSeSeCys), which contains a diselenide bond. Selenocystine units stabilize many selenoproteins with diselenide bridges. As a small molecular entity, selenocystine is assumed to take part in several stages of the body’s redox homeostasis pathways. This activity is similar to that of glutathione peroxidase. It was observed in a 10^−5^ mol/L solution of selenocystine using a GSH-Px assay [28] that selenocystine gets reduced to selenocysteine by glutathione reductase, indicating a pathway by which diselenides can also enter the glutathione peroxidase cycle. The presence of selenocystine indicates an increase in thioredoxin reductase activity [29,30]. Furthermore, selenocystine induces the ROS-mediated apoptosis of certain tumor cells [31,32,33].

Selenocysteamine (CysASeH) is of much less biological relevance, although it also exhibits glutathione peroxidase-like activity [28]. It is commonly used as a simple model compound to study the characteristics of the selenol moiety.

Selenocystamine (CysASeSeCysA) first became a compound of interest due to its antiviral activity [34]. Later, its apparent role in antioxidant pathways was revealed as a facilitator of glutathione oxidation [35]. The glutathione peroxidase-like activity of selenocysteamine–selenocystamine was utilized as a special coating on coronary artery stents. This facilitates nitric oxide production, thus lowering the risk of a possible subsequent infarction [36,37,38]. The structural formulae of the selenol-containing compounds in the main focus of this work are depicted in Figure 4.

## 4. Selenolate-Diselenide and Thiolate-Disulfide Redox Equilibria: Their Stepwise and Cumulative Macroscopic Constants

Glutathione is considered the “gold standard” in thiol-disulfide biochemistry. Hence, the redox behavior of every thiol-containing antioxidant is compared to glutathione. This standard procedure is aided by the fact that the species-specific physico-chemical acid-base [39] and redox [40,41] parameters of glutathione are characterized in detail in terms of their redox capacities. However, glutathione cannot easily reduce the diselenide bridge [42,43]. Based on our data, not even excessive amounts of glutathione are effective in reducing the diselenides to any significant amount. Thus, a more appropriate “standard” redox couple is needed for the investigation of selenolate-diselenide redox equilibria. Dithiothreitol (DTT), a stronger reducing agent, could be considered based on the literature recommendations [44,45]. Dithiothreitol undergoes a one-step, two-electron redox reaction, resulting in the formation of an intramolecular disulfide ring. The presence of an intermediate compound with DTT attached to another thiol during this redox reaction is negligible [46]. The main physico-chemical properties of DTT, such as log*K*_1_ and log*K*_2_, macroscopic thiolate protonation constants, and the E° redox potential, are known (log*K*_1_ = 10.1; log*K*_2_ = 9.2 [47]; *E*° = −0.323 V at pH = 7.0 [48]).

To obtain data describing selenolate-diselenide redox equilibria comparable to those of thiolate-disulfide systems, first, the glutathione–DTT redox reaction was investigated at a microspecies level [49].

The general stepwise scheme of selenolate−diselenide redox equilibria with DTT is depicted in Figure 5, where RSe^−^ and RSeSeR stand for the reduced and oxidized forms of the selenium-containing compounds, respectively.

Dithiotreitol is highly driven towards its oxidized, ring-closing form [47,50]. The reaction between diselenides and DTT can be reasonably treated in terms of *K*_C_, a cumulative equilibrium constant. This cumulative equilibrium constant is a conditional one and can be written as follows:(1)$$K_{C}=\frac{{\left[RSeH\right]}^{2}\cdot [{DTT}_{ox.}]}{\left[RSeSeR\right]\cdot \left[DTT\right]}$$

In the “selenol–diselenide” equilibria, the actually reduced form is the deprotonated selenol, i.e., the selenolate. All selenolates are basic moieties, and they take part both in acid-base and redox processes. These two types of reactions are interfering and co-dependent. To describe them in detail, it is necessary to define the protonation stage of each basic site. Microspecies are precisely defined species. Their protonation constants and equilibria are microconstants and micro-equilibria, respectively. The determination of all the related microscopic parameters is microspeciation. To elucidate the microscopic redox equilibrium constants, one has to consider the fact that selenolate (or thiolate) species participate in redox reactions, and many possible selenolate-bearing species exist for a certain compound. Consider selenocysteamine as a simple case: the selenolate-bearing species could be H_2_N-CH_2_-CH_2_-Se^−^ with a neighboring amino group or ^+^H_3_N-CH_2_-CH_2_-Se^−^ with a neighboring ammonium moiety. As an example, consider the microscopic redox equilibrium constant between H_2_N-CH_2_-CH_2_-Se^−^ (microspecies “a” in Figure 6) and the thiolate-bearing species of dithiothreitol (DTT^2−^). This microscopic constant, *k*^a^, refers to the reaction involving the “a” microspecies of CysASeH and the corresponding “a′” microspecies of CysASeSeCysA (Figure 6). Note that species “a” of the selenolate defines the diselenide species as “a′”, since “a′” is the only diselenide microspecies with protonation states in the vicinity of Se identical with “a”. Superscript “a” on *k*^a^ therefore unambiguously identifies all two microspecies in the microequilibrium in question. An example of this pH-independent, microscopic selenolate–diselenide equilibrium constant is in Equation (2):
(2)$$k^{a}=\frac{{\left[a\right]}^{2}\cdot [{DTT}_{ox.}]}{\left[a^{\prime }\right]\cdot \left[{DTT}^{2-}\right]}$$

The species-specific microconstant (*k*^a^) can be obtained as the product of the total concentration and the relative abundance of the respective microspecies, as seen in Equation (4). The relative abundance of the microspecies is a function of pH and the microscopic protonation constants. The concentration of the “a” CysASeH microspecies is expressed as:(3)$$\left[a\right]=\left[CysASeH\right]\cdot \chi_{a}=\left[CysASeH\right]\cdot \frac{1}{1+K_{1}\cdot \left[H^{+}\right]+K_{1}\cdot K_{2}\cdot {\left[H^{+}\right]}^{2}}$$
where [CysASeH] is the total solution concentration of CysASeH, *χ*_a_ is the relative abundance of microspecies “a,” *K*_1_ and *K*_2_ are the macroscopic protonation constants of CysASeH. Using analog equations such as Equation (3) for the concentrations of the other species involved in the redox reaction, one obtains:(4)$$k^{b}=\frac{{\left[a\right]}^{2}\cdot [{DTT}_{ox.}]}{\left[a^{\prime }\right]\cdot \left[{DTT}^{2-}\right]}=\frac{{\left[CysASeH\right]}^{2}{\cdot \chi }_{a}^{2}\cdot [{DTT}_{ox.}]}{[CysASeSeCysA]\cdot \chi_{a^{\prime }}\cdot [DTT]{\cdot \chi }_{{DTT}^{2-}}}=K_{C}\cdot \frac{\chi_{a}^{2}}{\chi_{a^{\prime }}\cdot \chi_{{DTT}^{2-}}}$$

Thus, if *K*_C_, the apparent equilibrium constant, is measured and the protonation constants together with the pH are known, all pH-independent, species-specific redox microconstants can be calculated.

## 5. The Role of pH in the Selenolate-Diselenide and Thiolate-Disulfide Redox Equilibria

Since redox and acid-base reactions of selenolate-containing compounds coexist, the observed redox equilibrium constant will be a pH-dependent apparent one. To get rid of the pH dependence and obtain a redox equilibrium constant that is pH-independent and characterizes a redox reaction between well-defined species, an appropriate method of determination has recently been introduced [40,41]. This method makes possible the determination of species-specific redox equilibrium constants (*k*^x^) and the species-specific, i.e., the standard redox potential (*E*°_x_), where “x” denotes the actual RSe^−^ microspecies in question; see Figure 4 and Figure 5. This method requires the knowledge of species-specific protonation constants, including those of the minor microspecies; otherwise, redox processes could only be characterized at the level of phenomena. The complete set of the microscopic protonation constants of the studied selenium-containing compounds (CysASeH, CysSeH) has recently been published [50].

### Selenocysteine and Selenocysteamine, Their Macroscopic and Microscopic Equilibria

The species-specific protonation equilibria of selenocysteamine and its diselenide, selenocystamine, are shown in Figure 6. The microspecies involved in redox processes are highlighted and color-coded. Figure 7 represents the same for selenocysteine and selenocystine. Note that the corresponding redox pairs have identical protonation states at the amino and carboxylate sites.

Macroscopic protonation equilibria indicate only the overall protonation state of the ligand with stepwise macroscopic protonation constants (*K_*1*_, K_*2*_,…*). In the microscopic protonation schemes, the microspecies (denoted with “a,” “b,” “c,” etc.) and the microscopic protonation constants (*k^N^, k^Se^,…*) are depicted. The superscript *k* indicates the group being protonated, while the subscript (if any) shows the site(s) already protonated. The indices of the protonation microconstants are not to be confused with the microscopic redox equilibrium constants. DTT^2−^ is only characterized at the macroscopic protonation level [47], since only its completely deprotonated form takes part in the redox reaction, and thus its concentration can be calculated using the macroscopic protonation constants.

The acid-base macro- and micro-equilibria are essential components of the evaluation method, as shown in Equations (3) and (4). Some protonation constant examples for selenocysteine are shown below:(5)$$K_{1}=\frac{\left[HL^{-}\right]}{\left[L^{2-}]\cdot [H^{+}\right]}$$
(6)$$\beta_{3}=K_{1}\cdot K_{2}\cdot K_{3}=\frac{\left[H_{3}L^{+}\right]}{[L^{2-}]\cdot {\left[H^{+}\right]}^{3}}$$
(7)$$k^{N}=\frac{\left[b\right]}{\left[a\right]\cdot \left[H^{+}\right]}$$

The concentrations of the various macrospecies comprise the sum of the concentration of those microspecies that contain the same number of protons, for example, in CysSeSeCys:(8)$$\left[H_{2}L\right]=\left[f^{\prime }\right]+\left[g^{\prime }\right]+\left[h^{\prime }\right]+\left[i^{\prime }\right]+\left[j^{\prime }\right]+[k^{\prime }]$$

For the selenolate-diselenide redox equilibria, the microspecies concentrations cannot be measured directly, so only the conditional equilibrium constants (*K*_C_) can be obtained directly. However, the pH-dependent apparent constants can be decomposed into pH-independent, species-specific equilibrium constants using Equation (4). Note that with the increasing number of neighboring basic sites next to the Se atom, the number of selenolate-bearing species, and thus the number of redox equilibria, increases exponentially. For example, the selenolate moiety in CysSeH is adjacent to two other basic sites: the amino and the carboxylate, and either one can be in two protonation states. Therefore, the number of selenolate-bearing microspecies is 2^2^ = 4. Accordingly, the selenolate moiety in the various CysSeH species exists in four different electrostatic environments, having thus four different oxidizabilities/redox potentials.

## 6. Experimental Techniques to Monitor Selenolate-Diselenide and Thiolate-Disulfide Redox Equilibria

For the determination of the conditional redox equilibrium constant (*K*_C_), the equilibrium concentrations of the compounds involved must be measured. In our studies, a ^1^H NMR spectroscopy was elaborated, but in principle, other analytical methods could be used, provided they selectively reflect the concentration of the various reactants and products in the solution. In practice, however, selenium-containing amino acids and DTT have no meaningful UV or fluorescence activity, and liquid chromatography cannot be used under certain pH conditions where the fast kinetics of the redox equilibrium [51] would perturb the concentrations in the case of chromatographic separation. Further advantages of NMR measurements include a relatively small sample requirement (sample volume 600 μL, concentration of analytes 1–5 mmol/L), biomimetic conditions (298 or 310 K, solvent H_2_O:D_2_O 95:5 (*v*/*v*), 0.15 mol/L ionic strength), and being able to discern signals of every compound involved (if not with 1D ^1^H NMR, then use of a 2D technique can be carried out, e.g. COSY, HSQC). For the NMR measurements, it is advised to use DSS (3-(trimethylsilyl)propane-1-sulfonate sodium) as a chemical shift reference and measure the pH with internal indicator molecules optimized for NMR [52,53]. The water resonance can be diminished by any preferred method, depending on the proximity of signals of interest to the water resonance.

It is recommended that stock solutions of the analytes be prepared in solvents that have been purged of air, preferably in a glovebox. The NMR measurements are best carried out immediately after preparing the sample solutions to diminish the risk of oxidation by air. To ensure that equilibrium had been reached, NMR spectra should be recorded immediately and after 1–2 h of sample preparation as well. For the quantitative analysis of NMR measurements, a peak fitting algorithm provides better results as opposed to the integration of overlapping signals. Using the chemical shift-pH profiles of the involved compounds (for the determination of their protonation constants) [50], the chemical shifts of the various signals at a given pH could be predicted, which aids in spectrum assignment and curve fitting. Since in Equation (1) the concentration units do not cancel out on the numerator and denominator sides, it is not sufficient to determine the relative concentration of the involved compounds from the NMR spectra. Additionally, the absolute concentrations of the selenium-containing compounds must be determined. Thus, a compound acting as a concentration standard must be utilized. The concentration standard is best chosen for a compound whose peak is always discernible and is preferably not a multiplet. Another highly advantageous feature of NMR spectroscopy lies in the fact that ^77^Se has an NMR active nucleus with ½ spin. Although the natural abundance of ^77^Se is 7.58% only, and it is usually considered an exotic nucleus, its chemical shift range is as wide as 2900 ppm, and its receptivity exceeds that of ^13^C. ^77^Se NMR spectra can directly reflect the status of Se, even in macromolecules [54]. These distinctive NMR properties make selenium redox research feasibility superior to related sulfur studies since NMR-wise the latter is a far less accessible element.

### 6.1. Conversion of Equilibrium Constants into Redox Potentials

The standard redox potential of dithiothreitol has to be inserted into the Nernst equation to obtain the redox potential values of the diselenide–selenolate redox pairs from their redox equilibrium constants involving DTT. Using a similar evaluation method as described above, we determined the pH-independent standard redox potential of DTT (specific for the species DTT_ox._/DTT^2−^) using the microscopic protonation constants [39] and standard redox potential [40] of glutathione microspecies. The species-specific standard redox potential was determined for the DTT_ox._/DTT^2−^ redox system as −0.403 ± 0.007 V.

In the knowledge of the selenolate–diselenide species-specific redox equilibrium constants and the standard redox potential of the DTT_ox._/DTT^2−^ redox couple, the species-specific standard redox potentials of the various selenolate-containing microspecies could be calculated using analogs equation to the one shown below with the example of CysASeH microspecies “a”.
(9)$${E{}^{\circ}}_{a}={E{}^{\circ}}_{{DTT}_{ox.}/{DTT}^{2-}}+\frac{R\cdot T}{z\cdot F}\cdot ln\frac{{\left[a\right]}^{2}\cdot [{DTT}_{ox.}]}{\left[a^{\prime }]\cdot [{DTT}^{2-}\right]}={E{}^{\circ}}_{{DTT}_{ox.}/{DTT}^{2-}}+\frac{R\cdot T}{z\cdot F}\cdot lnk^{a}$$

### 6.2. Correlation between Redox Potentials and Other Physico-Chemical Parameters

The species-specific redox potentials of diselenide-selenolate pairs are purified from pH effects, and they are therefore pH-independent, standard redox potential values. A remarkable feature of the data in Table 1 is that the redox potential values belonging to the same parent compounds (selenocysteine or selenocysteamine), but to different microspecies (a, b, c, d, f, or a, b), differ considerably. For example, the difference between the values of CysSeH microspecies “a” and “f” is 160 mV. This difference is purely a result of the different protonation states on the amino and carboxylate sites of CysSeH. Therefore, when observing apparent redox potentials that depend on pH, it has to be stated that the effect of pH on the apparent redox potential is two-fold: (1) primarily, the changing pH alters the protonation state of the selenolate moiety itself, thus changing the concentration of the redox active species; (2) secondarily, the changing pH modifies the protonation state and the concomitant electron densities in the neighboring moieties as well, thus modulating the redox propensity of the selenolate.

If the various selenolate-bearing species exhibit different standard redox potential values, it is straightforward to look for a correlation between these physico-chemical parameters and other parameters that are also highly determined by the electron density around the Se atom, e.g., selenolate-specific protonation constants, the nucleophilicity of the Se^−^ moiety, and the NMR chemical shift of the ^77^Se and its neighboring atoms. The relationship between the redox and acid-base parameters is elaborated in the next section. Here, we present the correlation between the standard redox potential values and ^1^H, ^13^C, and ^77^Se NMR chemical shifts determined for CysSeH species (Figure 8). This linear relationship is promising for predicting the redox characteristics of further selenocysteine derivatives based simply on their easily accessible NMR parameters. Note that this linear relationship is specific to selenocysteine species, i.e., derivatives of selenocysteine are expected to behave similarly, as seen in comparison with the case of cysteine-containing tripeptides [55]. This means that if the molecule’s selenolate-containing backbone is changed to, for example, selenocysteamine, selenohomocysteine, or any other compound, the linear relationship between NMR parameters and redox potential will have entirely different slope and intercept parameters.

## 7. Discussion: Comparison of Selenolate and Thiolate Redox Potentials

The close correlation between selenolate basicity and standard redox potentials (Figure 9) verifies the previous hypothesis that selenolate basicity and oxidizability are proportional properties, both being determined by the electron density environment of the Se atom. This correlation, as opposed to the one with NMR parameters, has the advantage of being independent of compound type. Both selenocysteine and selenocysteamine show the same linear regression between their species-specific protonation constants and redox potentials. It is also noteworthy that the regression line for the selenolate species is essentially parallel to the regression line of the thiolates, shifted downward by about 246 mV. This accentuates the fact that selenolates, apart from being less basic, are inherently stronger reducing agents than their thiolate counterparts.

### Selenocysteine Redox Data as Components to Interpret Selenoprotein Activities

The use of chemical shifts in protein NMR (including ^1^H, ^13^C, and even the few and exotic, but highly promising, ^77^Se spectra) can now serve as a sound means to predict selenolate oxidizability or diselenide reducibility/stability in proteins: a key parameter to understand and influence oxidative stress. It is often assumed that chemical shifts are highly susceptible to changes in the microenvironment of the NMR active nuclei. The correlation of chemical shift values with protonation constants and standard redox potentials could bring better knowledge about the chemistry and biological function of selenoproteins since, via the observed regression model, it is possible to estimate selenolate basicity and the concomitant standard redox potential. One obvious drawback of gathering species-specific NMR chemical shifts is the notoriously crammed ^1^H and 2D TOCSY spectra of proteins. This is why the measurement of ^77^Se chemical shifts in selenoproteins is so promising.

## 8. Conclusions

Quantification of the co-dependent redox and acid-base equilibria of the analogous selenolate-diselenide and thiolate-disulfide systems shows that selenols are inherently stronger antioxidants than thiols. Furthermore, reductions by selenols can take place under significantly more acidic circumstances, at a pH range some 3.5 units below that of their thiol counterparts. While it has long been known that the two-step selenol → diselenide and thiol → disulfide oxidation processes actually take place via the respective selenolate and thiolate forms, the new, species-specific redox potentials shed light on the fact that the reducing capabilities of all the multifunctional selenols and thiols are enhanced if the adjacent intramolecular basic sites (amino, carboxylate, etc.) are in their unprotonated forms. The correlations observed among redox potentials, basicities, and NMR chemical shifts establish sound predictability of hitherto inaccessible site-specific redox properties even in large molecules. This is especially and directly true for selenoproteins, where ^77^Se is an active, half-spin NMR nucleus, whereas thiolate-disulfide moieties are left to be characterized through ^1^H and ^13^C NMR chemical shifts near the sulfur atoms since sulfur has no useful NMR isotope.

## 9. Future Directions

The site- and species-specific redox potentials of the so far characterized small molecular selenolate–diselenide and thiolate–disulfide systems can be implemented into analogous moieties of large molecules. The relationships between NMR parameters and redox potentials make it possible to estimate the redox properties of specific parts of selenoproteins and sulfoproteins. This makes it possible to design and make selectively active antioxidants. 

## Figures and Tables

**Figure 1 antioxidants-12-00822-f001:**
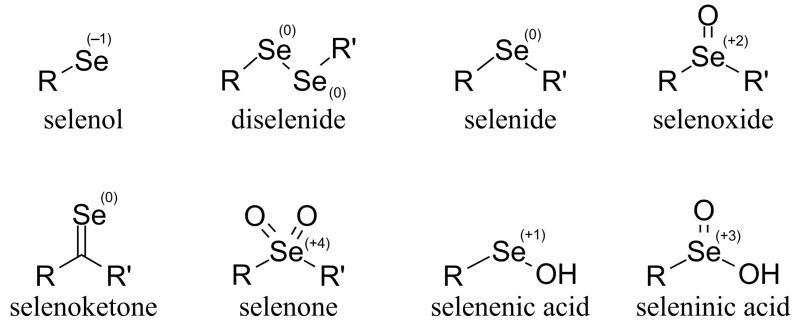
Some examples of organoselenium compounds, with the oxidation number of selenium in brackets.

**Figure 2 antioxidants-12-00822-f002:**
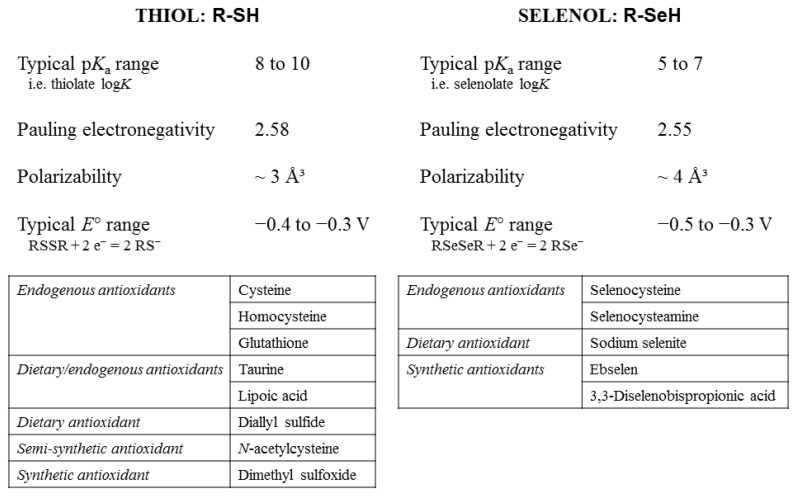
Some general physico-chemical properties of sulfur and selenium in thiol and selenol moieties, together with some examples of sulfur- and selenium-containing antioxidants (not necessarily confined to thiol/selenol-containing ones).

**Figure 3 antioxidants-12-00822-f003:**
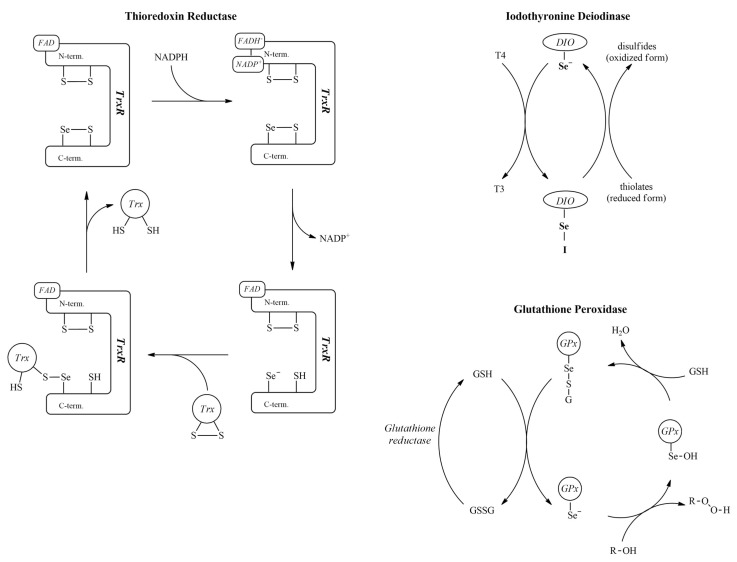
Catalytic mechanism of some selenoprotein enzymes: thioredoxin-dependent reduction using NADPH and FAD; redox cycle of glutathione peroxidase involving selenenic acid intermediate; deiodination mechanism of iodothyronine deiodinase.

**Figure 4 antioxidants-12-00822-f004:**
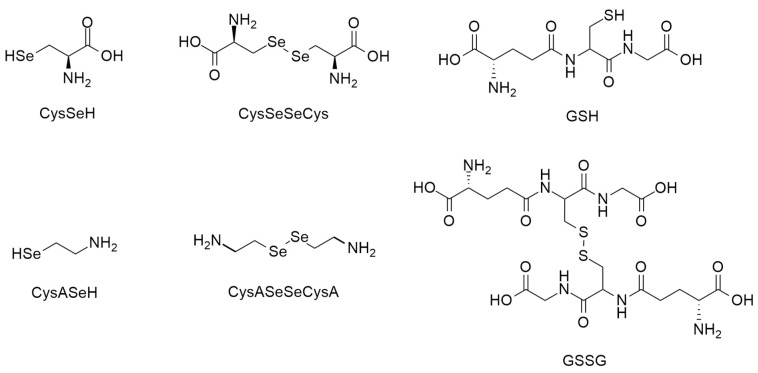
Constitutional formulae of selenocystine (CysSeH), selenocystine (CysSeSeCys), glutathione (GSH), oxidized glutathione (GSSG), selenocysteamine (CysASeH), and selenocystamine (CysASeSeCysA).

**Figure 5 antioxidants-12-00822-f005:**
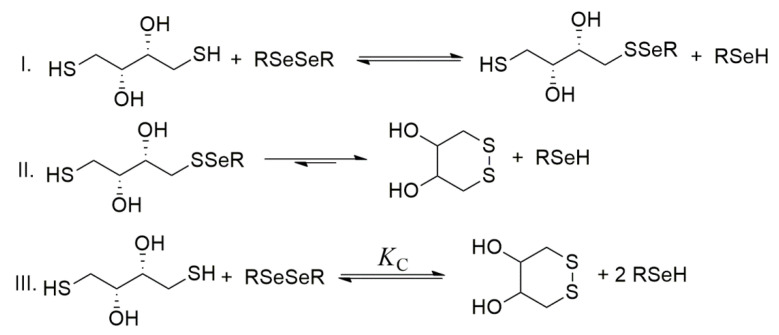
The two-step redox equilibrium between a diselenide compound and DTT^2−^ (reactions I and II), followed by their unified net reaction (III). Identical reaction schemes are applicable for disulfide analogs. The short arrow in reaction II indicates the low probability of the formation of the sulfur–selenium bridge.

**Figure 6 antioxidants-12-00822-f006:**
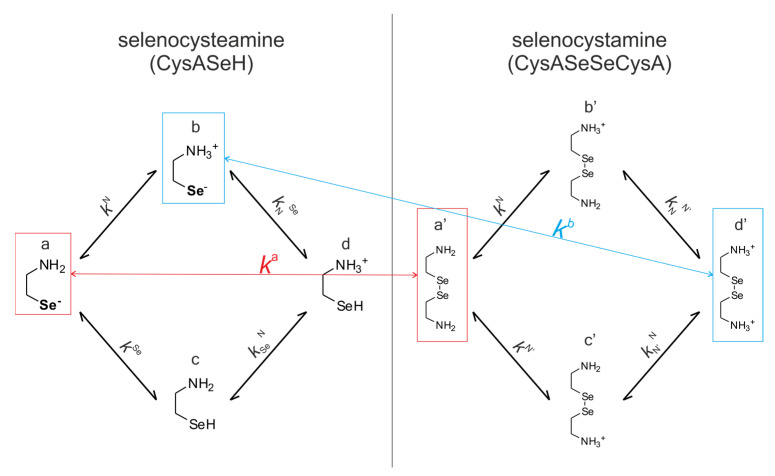
The coexisting protonation and redox microequilibria of selenocysteamine and selenocystamine; N, and Se labels denote the amino, and selenolate groups, respectively. Reproduced from [49].

**Figure 7 antioxidants-12-00822-f007:**
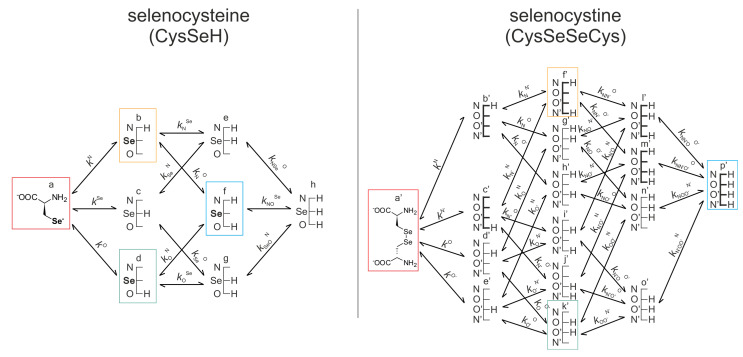
The protonation microequilibrium schemes of selenocysteine (**left**) and selenocystine (**right**). For simplicity, not every protonation microspecies are shown in a structural formula, rather a schematic structure depicts the microspecies with its basic sites where N, Se, and O labels denote the amino, selenolate, and carboxylate groups, respectively. The corresponding species participating in redox equilibria are bordered in color. Reproduced from [49].

**Figure 8 antioxidants-12-00822-f008:**
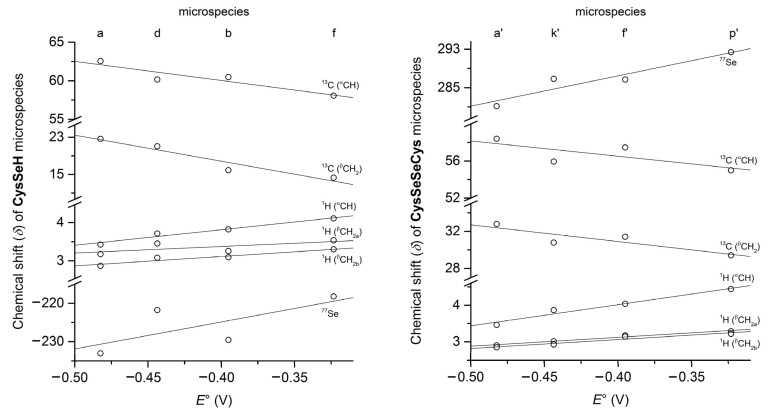
Correlation of selenocysteine standard redox potentials with the corresponding selenocysteine microspecies chemical shifts (**left**) and selenocystine microspecies chemical shifts (**right**). Reproduced from [49].

**Figure 9 antioxidants-12-00822-f009:**
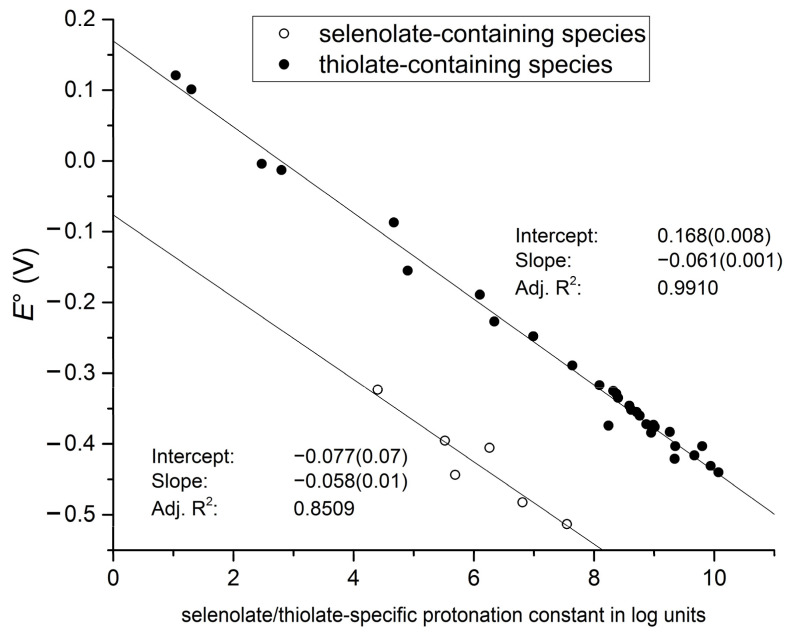
Correlation between standard redox potentials and species-specific thiolate protonation constants for various thiol-containing compounds (glutathione, cysteine, cysteamine, homocysteine, penicillamine, ovothiol A, DTT); the correlation between standard redox potentials and the species-specific selenolate protonation constants for various selenol-containing compounds (lower line, empty circles). Reproduced from [49].

**Table 1 antioxidants-12-00822-t001:** Standard redox potential values of selenocysteine and selenocysteamine, specified for their microspecies, along with their standard errors [49].

Selenocysteine		Selenocysteamine	
Microspecies	*E*° (V)	Microspecies	*E*° (V)
a	−0.482 ± 0.01	a	−0.513 ± 0.006
b	−0.395 ± 0.01	b	−0.405 ± 0.006
d	−0.444 ± 0.01		
f	−0.323 ± 0.01		

## Data Availability

The data presented in this study are available in the article.

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
