# Peer review of "Properties of Selenolate-Diselenide Redox Equilibria in View of Their Thiolate-Disulfide Counterparts"

_antioxidants, 2023, doi:10.3390/antiox12040822_

Round 1
Reviewer 1 Report
The manuscript described the selenolate-diselenide redox equilibria properties of their thiolate-disulfide counterparts. Generally, it is not very well fall in the scope of Antioxidants, please add more content related to the field of its antioxidative activity. For a review, it should comprehensively read a lot of literature and summarize it. There are only 40 references and based on the citation, it seems that the authors only focused on reference no. 34, which is rather narrow,
The purpose of the review should be provided. The summarized structure of the review should be displayed in the introduction as well, to make the whole structure of the manuscript more logical and readable
If there is a graph for how selenium affects the human body and what its antioxidative activity is in the body, that will be more clear and more straightforward.
I am unsure if some of the graphs are original or citations from the literature. If that is, please add the references. Please confirm Figures 2 and 3.
I am also unsure if some of the equations are calculated by the authors themselves or cited from the literature. If that is, please add the references. Some of the abbreviations are not clearly explained under the equations. Please confirm the equations 2-8
Author Response
The authors would like to thank Reviewer 1 for the insightful comments.
In general, in line with the recommendations of the Reviewer, the scope and material of the manuscript have been expanded, now with 56 literature references.
The Introduction section has been added, which outlines the purpose and the structure of the review.
The manuscript was expanded with a more thorough literature review of selenoprotein antioxidant activity, together with a fiture illustrating the proposed mechanisms.
Figures 2 and 3 are original, specifically created for this manuscript.
All the equations present in the manuscript are well-established in the field of physical chemistry and can be regarded as decendant from the ground-breaking work of Arrhenius, Bronsted and Bjerrum. The description of the symbols used in the equations can be found in the manuscript text.
Overall, the authors complied with all the suggestions of the Reviewer and would like to express gratitude for these recommendations that advanced the quality of the manuscript.
Reviewer 2 Report
This is a truely interesting and up=to=date reviewdevoted toselene, its derivates and way of studying, mailny using NMR-1H analysis. There are only limited remarks to be considered in the preparation of the final version of the manuscript.
1. Selene is a relative of sulfur, but sulfur is generally moreknown among biologists than selene. From the chemical point of view it would be justified to compare the main features of selene and sulfur (and the derivatives) somewhere in the beginning of the text.
2. A graphic representation of the main differences between the sulfur and selene components may supplement the text..
3. There are several found errors in English which suggest a thorough correction of the language. E.g. line 256 - :nucleur" instead of "nucleus", line 291 "elentron" instead of "electron", please do not use the bold character in line 202-204, unless it is fully confirmed. Etc.
Author Response
The authors would like to thank Reviewer 2 for the insightful comments and recommendations that advance the quality of the manuscript.
As per the recommendations, a Figure was added comparing the properties of sulfur and selenium, together with an expanded discussion in the text to aid understanding for biologists.
The found misspelling errors have been corrected.
Round 2
Reviewer 1 Report
The authors have addressed my comments.